# Selective RNA pseudouridinylation in situ by circular gRNAs in designer organelles

Lukas Schartel[1,2], Cosimo Jann[1,3], Anna Wierczeiko [4], Tamer Butto[5], Stefan Mündnich[5], Virginie Marchand[6], Yuri Motorin[7], Mark Helm [5], Susanne Gerber [4] & Edward A. Lemke [1,8] ✉

RNA modifications play a pivotal role in the regulation of RNA chemistry within cells. Several technologies have been developed with the goal of using RNA modifications to regulate cellular biochemistry selectively, but achieving selective and precise modifications remains a challenge. Here, we show that by using designer organelles, we can modify mRNA with pseudouridine in a highly selective and guide-RNA-dependent manner. We use designer organelles inspired by concepts of phase separation, a central tenet in developing artificial membraneless organelles in living mammalian cells. In addition, we use circular guide RNAs to markedly enhance the effectiveness of targeted pseudouridinylation. Our studies introduce spatial engineering through optimized RNA editing organelles (OREO) as a complementary tool for targeted RNA modification, providing new avenues to enhance RNA modification specificity.

RNA function is controlled by over hundreds of modifications[1]. While the tools for editing, writing, and analyzing RNA modifications are still evolving, they offer immense untapped potential for deepening our understanding of cellular functions controlled by RNA and advancing the field of synthetic biology.

One of the most abundant and functionally versatile RNA modification is pseudouridine (5-ribosyluracil, Ψ), an isomer of uridine. The structural differences to uridine, in particular the isomerization from a C–N glycosidic bond to a C–C bond, result in pseudouridine having an additional possibility to form a hydrogen bond and contribute to RNA stability[2]. In the cell, pseudouridines fulfill critical functions within a diverse array of small RNAs, ranging from regulating the function of rRNAs and stabilizing tRNAs to regulating splicing within small nuclear RNAs[3]. In the context of in vitro-transcribed mRNAs, pseudouridine plays a pivotal role as a potent enhancer of translation[4]. Additionally, when situated within stop codons, it possesses the unique ability to mediate stop-codon suppression[4,5]. Applications of engineering pseudouridine into RNAs focus on modifying stop codons within mRNAs, which represents an exciting opportunity to target many heritable diseases caused by premature stop codons, such as variants of cystic fibrosis[6]. These methods either rely on endogenous factors limiting their efficiency[7] or overexpression of pseudouridine synthases that could potentially cause unwanted off-target effects[8].

Cells are equipped with numerous enzymes capable of synthesizing pseudouridine in RNAs. Most of these enzymes function as standalone pseudouridine synthases, each exhibiting specific affinities for particular structural features[9]. We used the RNA-guided pseudouridine synthase dyskerin (DKC1) to construct an RNA-modifying organelle that maximizes target versatility. H/ACA box small nucleolar RNAs (snoRNAs) form complexes with DKC1 to function as guide RNAs (gRNAs). These H/ACA box RNAs possess a common secondary structure with characteristic pockets harboring antisense guide sequences[10]. These pseudouridinylation pockets can be engineered to complement any target RNA, making gRNA-dependent DKC1 the ideal candidate to be used in a programmable pseudouridine-dependent RNA-modifying organelle[7].

Here, we demonstrate that a film-like organelle architecture enables mRNA-selective pseudouridinylation as a proof-of-principle

[1]Biocenter, Johannes Gutenberg University, Mainz, Germany. [2]Departments of Biology and Chemistry, IMPRS on Cellular Biophysics, Mainz, Germany. [3]Institute of Molecular Biology (IMB) Postdoc Programme, Mainz, Germany. [4]Institute of Human Genetics, University Medical Center of the Johannes Gutenberg University, Mainz, Germany. [5]Institute of Pharmaceutical and Biomedical Sciences, Johannes Gutenberg-University Mainz, Mainz, Germany. [6]Université de Lorraine CNRS, INSERM, UAR2008/US40 IBSLor, EpiRNA-Seq Core Facility, Nancy, France. [7]Université de Lorraine CNRS, UMR7365 IMoPA, Nancy, France. [8]Institute of Molecular Biology, Mainz, Germany. ✉e-mail: edlemke@uni-mainz.de

for a particular mRNA-modifying organelle. The concept behind film-like organelles is to create a unique biochemical environment within a living mammalian cell by borrowing principles from 2D phase separation, where the film-like structure still stays in exchange with the cytoplasm[11,12]. Primarily, mRNAs targeted at the thin film should be selectively modified, whereas mRNA located elsewhere in the cytoplasm should get less frequently modified. As the environment of the thin film is in equilibrium with the cytoplasm, there are no physical boundaries to be passed, which abrogates the need for complex transport machinery. Furthermore, components can be shared between the host cell and the designer organelle so that film-like organelles can be simple in design and built from a few critical components while borrowing others from the cytoplasm.

We also show that mRNA pseudouridinylation can be significantly enhanced using circular gRNAs. Our results show the functionalities of film-like organelles for RNA editing and highlight the power of circular gRNAs for targeted stop-codon suppression via pseudouridine, offering new possibilities ranging from synthetic biology to RNA therapeutics.

## Results

The unique ability of pseudouridine to function as a stop-codon suppressor allows us to design a simple mRNA reporter encoding for a fluorescent protein with a premature stop codon. When stop-codon suppression occurs, the mRNA produces a fluorescent protein that can be detected using flow cytometry. For this purpose, we designed a gRNA derived from H/ACA box RNA pugU2-34/44 of *Xenopus laevis*, which has recently been used to mediate targeted stop-codon suppression[13]. We customized the base pairs within the pseudouridinylation pocket of the gRNA to target an iRFP-eGFP[39TAG] reporter, which should produce eGFP only if pseudouridinylation of the premature stop codon occurs.

We built upon our previous design of orthogonally translating film-like organelles[12], which consist of an N-terminal domain of the rodent LCK tyrosine kinase (LCK[1-10], which we from here on refer to as LCK), serving as a plasma membrane (PM) anchor (Fig. 1) an intrinsically disordered region (IDR, aa 2–267) derived from the protein FUS, and major capsid protein (MCP), which serves as a recruitment domain. These parts and various control constructs lacking either the LCK, IDR, or MCP domain were N-terminally fused with dyskerin under the control of a CMV promoter. We utilized a truncated version of dyskerin (aa 22–424), which we refer to hereafter as DKC1. This version lacks NLS/NoLS signals at the N or C termini and we added a nuclear export signal (NES) at the N terminus, which is present in all DKC1

fusion constructs throughout this study. It has also recently been shown that only isoform 3 of dyskerin, which is N/C-terminally truncated, can mediate efficient pseudouridinylation of an mRNA target when overexpressed[8]. Airyscan resolution images of a DKC1 fusion construct highlight the differential localization at the PM when fused to LCK (Fig. 2a and Supplementary Fig. 1).

To first assess if such complex fusion constructs can still mediate stop-codon suppression through pseudouridinylation, we used an iRFP-eGFP[39TAG] reporter harboring two RNA stem loops derived from the ms2 phage in the 3′ UTR, which we term the efficiency reporter (Fig. 2b). These RNA loops bind specifically to the MCP domain present in our organelle constructs, allowing selective recruitment of our efficiency reporter. We then transiently transfected HEK293T cells with our efficiency reporter, as well as different constructs of DKC1 and a custom-designed gRNA under the control of a U6 promoter. Next, we analyzed our cells with flow cytometry (Supplementary Fig. 2) and calculated the percentage of GFP-positive cells, indicating successful stop-codon pseudouridinylation. A LCK-FUS[2−267]-MCP-DKC1 fusion construct in which the efficiency reporter is actively recruited via the ms2/MCP tagging system mediated efficient nonsense suppression when co-transfected with a gRNA, albeit with lower efficiency than NES-DKC1. LCK-DKC1 constructs missing either the IDR or MCP showed lower efficiency in comparison (Fig. 2b, c).

Having established the functionality of our RNA editing organelle (REO) construct, we investigated whether REOs can mediate selective stop codon pseudouridinylation. For this purpose, we switched to a bidirectional expression plasmid encoding mCherry[190TAG_2xms2] and eGFP[39TAG], where both reporters are separately transcribed under the control of a CMV promoter, which we refer to as the selectivity reporter. We then transfected HEK293T cells with our organelle construct and two specific gRNAs targeting the premature stop codons in our fluorescent reporters. In this scenario, we expect mCherry fluorescence to dominate in cells that are co-transfected with our REO because the mCherry reporter mRNA is actively recruited via the ms2/MCP tagging system. In the flow cytometric analysis, we calculated the relative fold change of selectivity, which we define as the ratio of the median mCherry and eGFP intensities. Indeed, we observed a greater than 2.5-fold increase in selectivity in cells hosting our designer organelle compared to the cells expressing NES-DKC1 (Fig. 2d, e). This effect is dependent on ms2 loops as a selectivity reporter without ms2 loops cannot be selectively pseudouridinylated (Supplementary Fig. 3). Conversely, inverting the selectivity reporter design (eGFP[39TAG_2xms2]/mCherry) qualitatively inverted this effect (Supplementary Fig. 4).

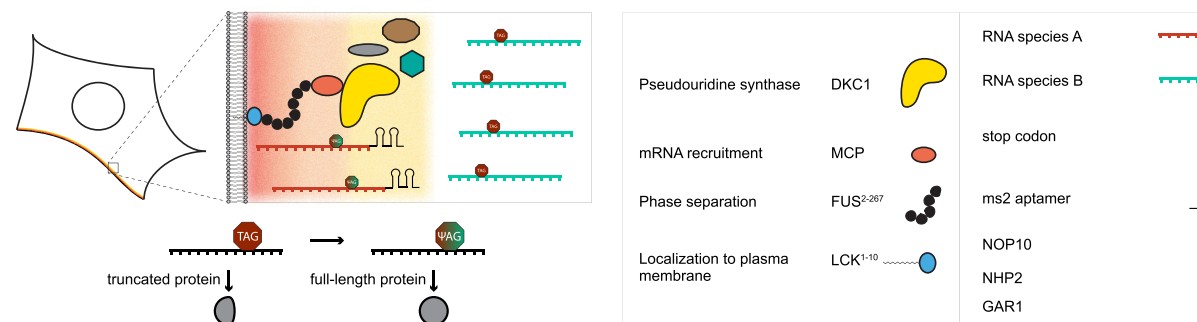

**Fig. 1 | Film-like organelles for selective RNA modification.** Cartoon showing a film-like organelle mounted on the plasma membrane. The colored shading indicates the organelle microenvironment, RNA is recruited through ms2 loops into the organelle, and modified stop codons result in the full-length production of protein. gRNAs are not shown to keep the illustration simple. DKC1 is active as a tetrameric complex with NOP10, NHP2, and GAR1, which are also drawn here for reference. Our method enabled us to utilize the endogenous pool of those, which alleviated the need to engineer any of those.

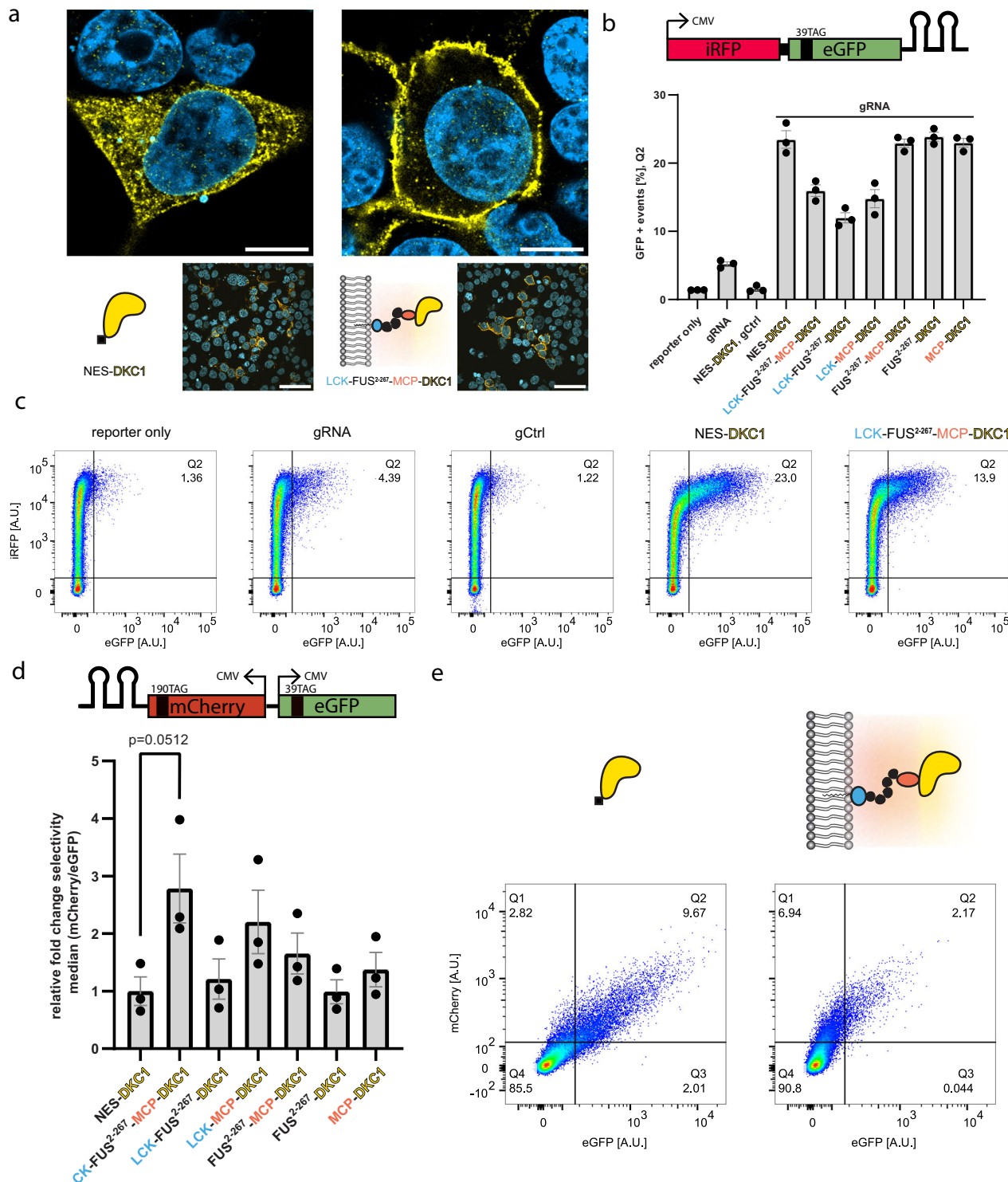

### Circular gRNAs markedly enhanced pseudouridine-mediated stop-codon suppression

To optimize the efficiency of our organelle for pseudouridine-dependent stop-codon suppression, we were inspired by an elegant approach for the circularization of small aptamers[14]. We chose to express our gRNA that targets the efficiency reporter (Fig. 2b) under the control of a U6 promoter flanked by twister ribozymes and a 10-nt polyadenine linker (Fig. 3a). After expression, twister ribozymes cleave themselves, leaving an RNA stem that is ligated by the endogenous RNA ligase RtcB. Excitingly, circular gRNAs markedly increased the

population of GFP-positive cells, allowing up to 10% more GFP-expressing cells in the case of our organelle (Fig. 3b, c). Circularization of our gRNAs was confirmed by RT-PCR with inverse primer pairs where only a circular gRNA results in a PCR product (Fig. 3d). We suspect that the increase arises from the greater resistance of circular gRNAs to exonucleases, allowing them to accumulate at much higher levels than their linear counterparts as it has been shown in the past for circular aptamers and circular RNAs for ADAR mediated RNA editing[14–16]. In parallel, a recent study has demonstrated that circular RNAs are preferentially exported from the nucleus[17]. This finding offers

**Fig. 2 | Film-like organelles mediate stop-codon suppression by selective pseudouridinylation. a** Immunofluorescence of HEK293T cells transfected with NES-DKC1 (left) and LCK-FUS$^{2-267}$-MCP-DKC1 (right), alongside a cartoon depicting DKC1 (yellow), MCP (pink), FUS$^{2-267}$ (black), and LCK (blue). Cells were stained against a FLAG tag with Alexa-488; the nuclei were stained with DAPI. Scale bars = 10/50 μm (top/bottom images). Top images have been taken with an LSM900 Airyscan2, bottom images have been taken with an BC43 spinning disk confocal microscope. Imaging was repeated once with similar results. **b** Pseudouridine-mediated stop-codon suppression was measured by co-transfecting HEK293T cells with an iRFP-GFP$^{39TAG\_2xms2}$ reporter, a specific gRNA targeting eGFP, and various fusion constructs of DKC1. 5 × 10$^4$ live cells were analyzed using flow cytometry and GFP + events [%] were plotted. A control gRNA (gCtrl), in which target sequences have been randomized, served as a control. The

standard error of the mean (SEM) from three independent experiments are shown. **c** Representative flow cytometry scatter plots shown for key samples in (**b**), Q2 indicates the eGFP positive population. **d** A reporter plasmid carrying mCherry$^{190TAG\_2xms2}$ and eGFP$^{39TAG}$ under the control of two independent CMV promoters served as a selectivity reporter. HEK293T cells were co-transfected with the selectivity reporter as well as two specific gRNAs targeting mCherry/eGFP (both transcribed from the same plasmid, each under the control of a U6 promoter) and various fusion constructs of DKC1. 5 × 10$^4$ live HEK293T cells were analyzed using flow cytometry. The relative fold change in selectivity, which is defined as the ratio of median mCherry/eGFP intensities, was normalized to NES-DKC1 and plotted. The SEMs from three independent experiments are shown. An unpaired two-sided *t*-test was performed (*n* = 3; **p* < 0.05) (**e**) Representative flow cytometry scatter plots for data displayed in (**d**). Source data are provided as a Source Data file.

another potential explanation for the observed higher pseudouridine modification efficiencies with circular gRNAs since all our DKC1 constructs are located in the cytoplasm.

We speculated that circular gRNAs could be further engineered and perhaps also actively recruited to our REOs. To test this, we added different versions of ms2 loops 5' and 3' of our gRNAs (Supplementary Fig. 5). In parallel, we designed LCK-FUS$^{2-267}$-4xλN22-FUS$^{2-267}$-MCP-DKC1 organelles, which comprise four repeats of the λN22 peptide. Similar to the ms2/MCP system, λN22 peptides facilitate the recruitment of RNAs tagged with boxB RNA loops[18]. In addition, we added another FUS$^{2-267}$ linker between the two RNA-binding proteins, hoping the construct would retain flexibility and allow full access to both recruitment domains. Because we tagged our circular gRNAs with ms2 loops, we switched to a bidirectional mCherry$^{190TAG\_4xboxB}$/eGFP$^{39TAG}$ reporter to avoid competition between circular gRNAs and the reporter for binding MCP. After comparing cells co-transfected with ms2 tagged to unlabeled circular gRNAs, we observed a further optimization in selectivity up to 4-fold when using these gRNA-optimized RNA editing organelles (OREO).

To establish the generality of our approach, we also tested three more previously used membrane anchors[12] to form OREOs (Supplementary Figs. 6 and 7). We show data for additional Golgi, ER, and mitochondrial-targeted OREOs that perform similarly to the LCK, i.e., the PM anchoring organelles. In all cases, we observed a beneficial effect of inserting the FUS$^{2-267}$ LCD in our constructs.

### Selective pseudouridinylation with circular gRNAs

Having established the enhancing effect of circular gRNAs, we tested their effectiveness in our dual-color reporter experiment introduced above in Fig. 2d. We transfected HEK293T cells with our selectivity reporter and circular gRNAs for mCherry$^{190TAG\_2xms2}$ and eGFP$^{39TAG}$, as well as our OREO. We observed that cells harboring organelles maintained high selectivity (Fig. 4a), preferentially producing mCherry. The enhancing effect of circular gRNAs was more apparent when we looked at the percentage of mCherry-positive cells, which was, on average, more than doubled compared to REOs that use linear gRNAs (Fig. 4b, c).

Direct methods to validate RNA pseudouridinylation are still a contemporary research field[19,20]. To validate RNA pseudouridinylation qualitatively but directly, we chose to use direct RNA-sequencing (DRS; Oxford Nanopore Technologies). DRS allows natural RNA to be sequenced without prior amplification steps. Specific current changes during the sequencing process can then be detected and used to identify the underlying canonical bases. It has been shown that DNA or RNA base modifications can differ in their current signal pattern from their canonical counterparts, allowing those modifications to be identified[20,21]. The most commonly observed pattern for pseudouridine modifications is a systematic uridine−cytosine mismatch, i.e., pseudouridine is detected as a C[22,23]. This facilitates the classification of individual reads into unmodified and modified reads based on basecalling error information, allowing quantification of pseudouridine modifications at specific sites. We verified the applicability of DRS by sequencing fully modified

synthetic RNA oligos, mimicking our target sequences within mCherry and eGFP (Supplementary Figs. 8, 10, and 11). We extracted poly-A RNA from cells used in our selectivity assay (Fig. 4c) and from cells in which a non-targeting circular gRNA was expressed with our organelle and selectivity reporter, followed by the direct RNA-sequencing protocol from Oxford Nanopore Technologies. After mapping the base called reads onto the eGFP and mCherry reporter sequences, the basecalling error patterns, including U−C mismatches, were extracted for all reads mapping on the mCherry and EGFP reporter sequence from both the NES-DKC1- and designer organelle-transfected cells. Relative changes are consistent with our flow cytometry data; that is, OREO-targeted mCherry is preferentially pseudouridinylated at the desired position with mCherry/eGFP fold changes increasing from 2.9-fold to 25.1-fold (Fig. 4d, top) (Supplementary Fig. 9).

In parallel, we employed BID-seq (bisulfite-induced deletion sequencing) to cross-validate our DRS results. BID-seq allowed quantification of pseudouridine at single base resolution and was also able to confirm the observed modification trend albeit with lower overall fold changes (mCherry/eGFP ratios increasing from 3.4-fold to 15.7-fold) (Fig. 4d, bottom) as observed in DRS.

### OREOs modify disease-relevant mRNA

To test OREOs beyond our standard fluorescent reporters, we designed circular gRNAs for AldoB$^{148TGA}$, a mutation that renders the AldoB gene non-functional, causing hereditary fructose intolerance. We now expressed from a bidirectional promoter the AldoB(148TGA) mutant gene fused to ms2 loops as well as the eGFP gene to track off-target effects. Western blot analysis shows an efficient high on-target modification leading to expression of full-length AldoB gene while substantially reducing off-target eGFP levels (Fig. 4e, f).

## Discussion

One of the challenges of synthetic biology is to engineer biochemical reactions with high specificity in complex cellular systems. Our study contributes to this challenge by showing that artificial film-like organelles can mediate selective RNA modifications. Our designer organelle approach allows the spatial engineering of pseudouridine, thereby lowering the chance for unwanted off-target modifications. This was demonstrated using the dual-color reporter (Fig. 1), where full-length GFP is made when STOP codons get altered in the cytoplasm, while full-length mCherry is made when the OREO alters the STOP codon.

Additionally, by using circular gRNAs, our organelles show increased levels of pseudouridinylation indicated by pseudouridine-mediated stop-codon suppression. Because pseudouridines also influence other cellular processes, such as splicing, applying the OREO concept to the regulation of splicing might be attractive. We believe this designer organelle approach to engineering RNA modifications can be broadened to encompass other RNA modifications that state-of-the-art assays can accurately detect.

We show that using a ribozyme-mediated circularization approach can enhance the efficiency of RNA-guided

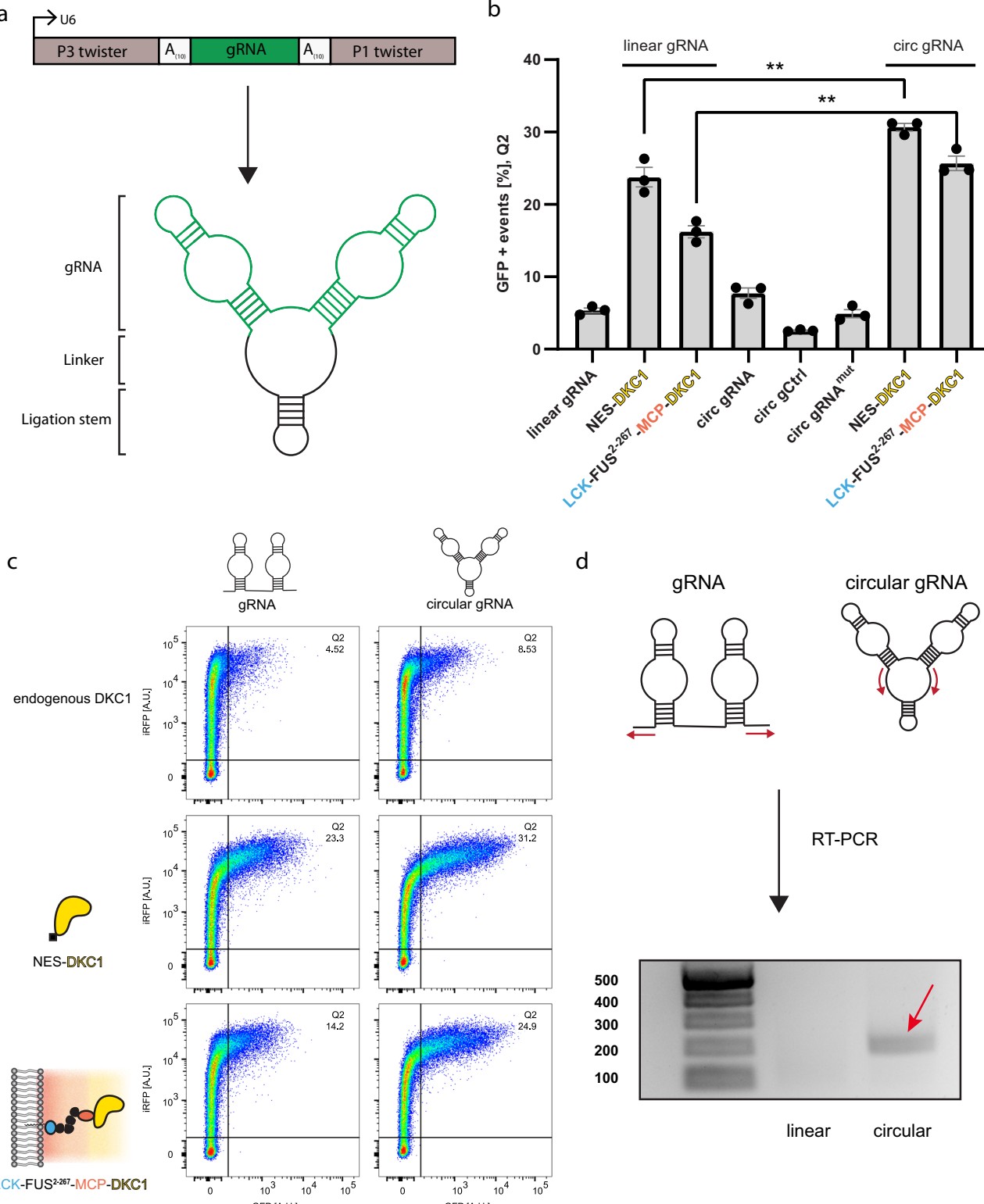

**Fig. 3 | Circular gRNAs allow enhanced pseudouridine-mediated stop-codon suppression. a** Cartoon depicting the generation of circular gRNAs from a U6 promoter using the Tornado system. A 10-nt polyadenine linker was incorporated 5′ and 3′ of the gRNAs to maintain flexibility. **b** Circular gRNAs outperform linear gRNAs in HEK293T cells co-transfected with the efficiency reporter (Fig. 1b) and DKC1 constructs. The SEMs of three independent measurements are shown, in which $5 \times 10^4$ live cells were measured using flow cytometry, an unpaired two-sided

$t$-test was performed ($n = 3$; $p = 0.0089$ for NES-DKC1, $p = 0.0019$ for LCK-FUS$^{2-267}$-MCP-DKC1, $^{**}p < 0.01$). **c** Flow cytometry scatter plots shown for key samples in (**b**). **d** circularization of gRNAs was confirmed by RT-PCR using inverse primers, which only result in a PCR product if the gRNA is circularized. The expected product size is 199 bp. Purified PCR samples were run on an agarose gel (ladder is given in base pair units). RT-PCR was repeated once with similar results. Source data are provided in Source Data file.

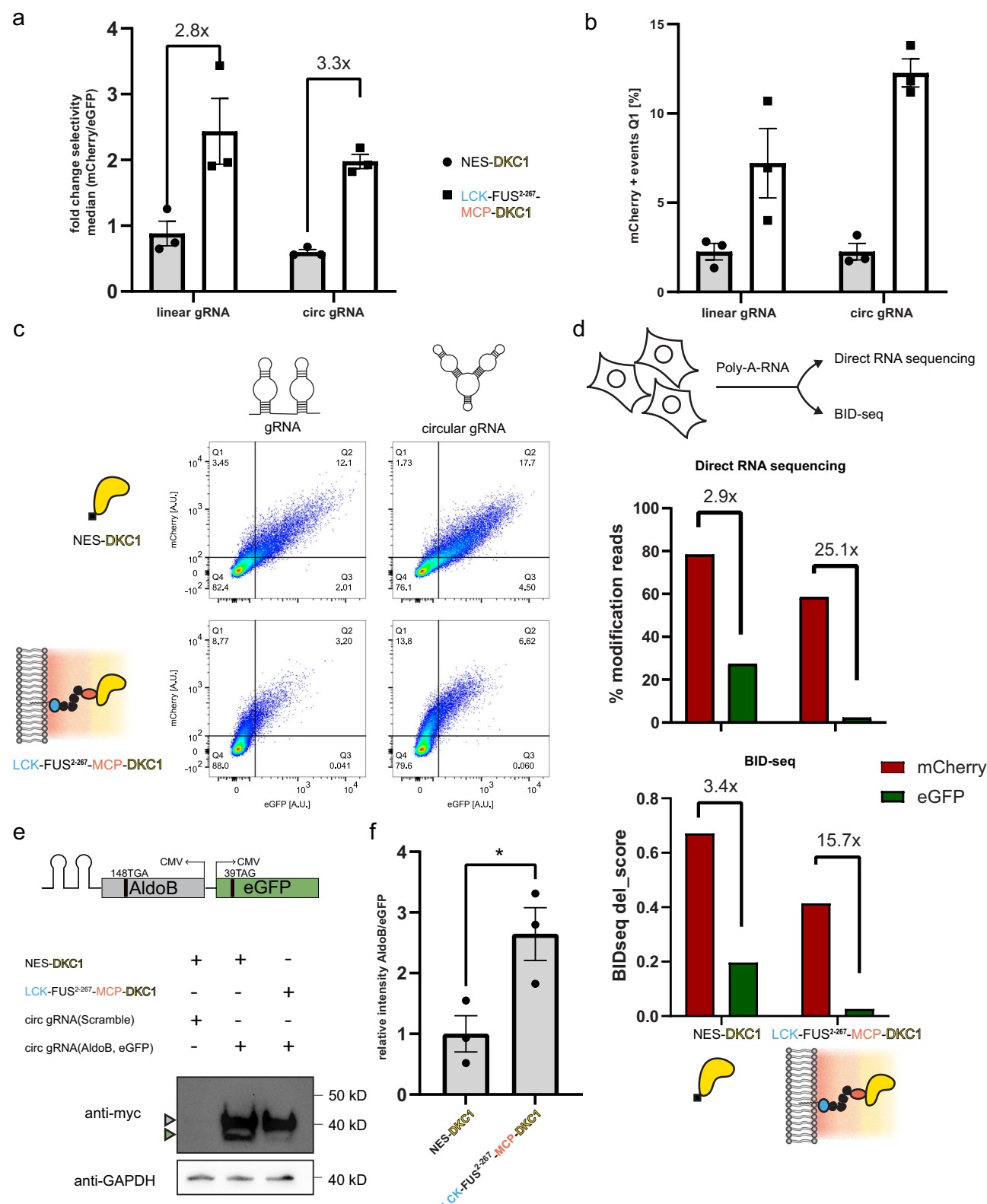

pseudouridinylation. Through further optimization of linker sequences and RNA folding kinetics, circular H/ACA box RNAs can possibly be optimized to mediate even higher rates of pseudouridinylation. Additionally, circular gRNAs can potentially be further functionalized with RNA loops, allowing active recruitment into our designer organelles.

More than 10% of genetic diseases are caused by premature stop codons. Stop codon suppression by pseudouridine offers great potential in treating these conditions[24]. The enhancing efficiency of circularized H/ACA box-derived snoRNAs holds the potential for targeting such diseases in the future. In a recent study, overexpression of near cognate tRNAs has been shown to increase pseudouridine-mediated stop codon suppression. Our enhanced pseudouridinylation system, developed through this approach, holds the potential to synergize with near-cognate tRNAs, further facilitating enhanced stop codon suppression by pseudouridine. Currently, our ability to detect

**Fig. 4 | Circular gRNAs increase designer organelle-mediated pseudouridinylation. a** HEK293T cells were co-transfected with the selectivity reporter (Fig. 1c), gRNAs, and either NES_DKC1 or LCK_FUS[2-267]_MCP_DKC1.5 × 10[4] live cells were measured and the fold change in selectivity was calculated for linear and circular gRNAs. **b** Simultaneously, cells were gated and analyzed for mCherry + cells Q1, as depicted in (**c**). SEMs of three independent replicates are shown for (**a, b**). **c** Representative flow cytometry scatter plots shown for (**a, b**). **d** Poly-A RNA was extracted from transfected HEK293T cells and analyzed via direct RNA sequencing using nanopore sequencing via BID-seq. The percentage of modified reads was plotted and the fold change between reporter modifications was compared. Analogous modifications from BID-seq levels (BIDseq del_score: 0–1 refers to unmodified and fully modified, respectively) were compared between NES-DKC1 and

OREO transfected cell reporters. **e** Selective pseudouridinylation of AldoB(148TGA) expressed from a modified selectivity reporter with both proteins tagged with 2× myc tag on the c-terminus. Transfected HEK293T cells were lysed and enriched for full-length AldoB and eGFP via myc-trap beads. A representative western blot of the eluted proteins is shown(top), triangles (gray/AldoB; green/eGFP) indicate the respective protein bands. A small aliquot of lysate was used for myc-trap pull-down and used as a loading control (bottom). Unprocessed blots are available in Source Data file. **f** Protein bands were quantified and the relative fold change in AldoB/eGFP intensities was calculated. Three independent experiments were performed. An unpaired two-sided $t$-test was performed ($n = 3$; $p = 0.0359$, *$p < 0.05$) and SEM was plotted. Source data are provided as a Source Data file.

RNA modifications makes transcriptome-wide detection of off-target effects a challenge on its own. Nevertheless, we can show that when overexpressing a disease-relevant gene with an "off-target" gene, we can selectively modify our target RNA using OREOs. The use of our reporter design enables us to visualize how selectivity for RNA editing can be achieved when using film-like OREOs, which greatly expands the use of such technologies for enzymatic engineering in general. Specificity and limited off-target effects will always be the principal reasons for selecting the optimal modification tool. Because the film-like organelle is accessible, i.e., open, from the cytoplasm, a major benefit remains the simplicity, as also, in this case, only one of the four components of the DKC1 machinery had to be spatially concentrated to the organelle. At the same time, such film-like systems are in equilibrium with the cytoplasm, and thus, biomolecules can still diffuse out of the OREO and into the cytoplasm, and an infinitely high selectivity cannot be expected.

Previous orthogonally translating organelles (OTOs), have been used to alter the genetic code of translation to introduce noncanonical amino acids into proteins, and selectivity increases of ~10 fold have been found[11]. The "open" architecture of an OREO or an OTO might even enable their combination to reprogram RNA function and/or protein synthesis holistically beyond current single methods. This open architecture is inspired by concepts from phase separation, where organelle-like structures in equilibrium with the cytoplasm can form. We also note that even without using the FUS LCD, all our organelles outcompete the cytoplasmic distributed NES-DKC1 system in terms of selectivity. However, understanding such highly concentrated 2D systems' biophysical nature is still part of contemporary research even for single protein systems[25,26]. Thus, here, we focus on the engineered function and the membrane-targeted appearance when referring to a film-like OREO.

The open design might also facilitate combining film-like organelle technology with other clever and elegant methods, such as CRISPR and ADAR-based RNA engineering[27,28]. OREOs thus do not compete with existing methods but might provide another layer of tuning specificity. Thus, the combination of tools might realize true off-target-free RNA editing in the future.

## Methods
### Cloning
Constructs in this work were cloned using either restriction cloning, golden gate assembly, or Gibson assembly. The efficiency reporter was cloned into a pCI vector (E1731, Promega). The dual fluorescent protein reporter (selectivity reporter) was cloned in a pBI-CMV1 vector (Clontech 631630), with ms2 tagged fluorescent protein (mRNA) version in one multiple cloning site and ms2 free version in the other[11]. The DKC1 constructs were assembled in pcDNA3.1 backbone via Gibson assembly. gRNAs were constructed by amplifying forward and reverse primers and cloned into a puc57 backbone under the control of a U6 promoter using a golden gate assembly. Circular gRNAs were cloned into a puc57 backbone containing P3 twister and P1 twister ribozymes using golden gate assembly.

### Cell culture
HEK293T cells (ATCC CRL-3216) were maintained in Dulbecco's modified Eagle's medium (DMEM, Gibco 41965-039) supplemented with 1% penicillin-streptomycin (Sigma-Aldrich P0781), 1% L-glutamine (Sigma-Aldrich G7513), 1% sodium pyruvate (Life Technologies 11360), and 10% FBS (Sigma-Aldrich F7524). Cells were cultured at 37 °C in a 5% $CO_2$ atmosphere and passaged every 2–3 days up to 20 passages.

### Flow cytometry
$1.1 \times 10^5$ HEK293T cells were seeded in 24well plates and transfected roughly 17 h later using jetPrime (Polyplus) as indicated by the manufacturer. Plasmids for organelle/gRNA/reporter constructs were transfected in a 1:1:1 ratio for experiments in Fig. 1b with a total of 1.2 μg DNA per well. The media was exchanged after 4 h and cells were grown for additional 48 h. For flow cytometry measurements cells were washed once with PBS and trypsinized. Cells were resuspended in 900 μl resuspension buffer 1 (1× PBS, 10% FBS, 2 mM sodium azide, 2 mM EDTA) and centrifuged for 5 min at $400 \times g$ at 4 °C. Supernatant was poured off and cells were resuspended in 900 μl resuspension buffer 2 (1× PBS, 3% BSA, 2 mM sodium azide, 2 mM EDTA) and centrifuged again for 5 min at $400 \times g$ and 4 °C. Supernatant was again poured off and cells were resuspended in 300 μl resuspension buffer 2 and cooled on ice until the measurement. Data was collected with a LSRFortessa SORP (BD Biosciences). DAPI (50 μg/mL) was added before measurement to distinguish live from dead cells. $1.1 \times 10^5$ live cells were measured for each sample. Data was analyzed using FlowJo (v10.7.1.) HEK293T cells were distinguished based on FSC-A and SSC-A. Single cells were gated based on SSC-W and SSC-A. Live cells were distinguished based on SSC-W and 405–450/50 channels. For experiments using the efficiency reporter cells were then divided in four quadrants based on the reporter-only control (Supplementary Fig. 2). To estimate transfection efficiency from iRFP expressing cells, cells were gated based on SSC-F and 640–730/45 channel. Final GFP + percentages (Q2) were calculated by normalizing to iRFP + percentages of NES-DKC1. Normalized GFP + percentages (Q2) have been plotted in Figs. 1b and 3c. Data were plotted using GraphPad Prism software (v9.1.1). For experiments using the selectivity reporter eGFP fluorescence was acquired using a 488 nm laser and a 530/30 bandpass filter and for mCherry, a 561 nm laser with a 610/20 bandpass filter. Cells were gated similar to efficiency reporter experiments and quadrants were adjusted to a reporter-only control in Q4. For analysis of selectivity experiments a NOT gate for Q4 was created in FlowJo to collect total eGFP and mCherry intensities from Q1 to Q3. Relative fold changes in selectivity were calculated by dividing median mCherry intensities by median eGFP intensities. Calculated fold changes were normalized to NES-DKC1.

### Microscopy
$3 \times 10^4$ HEK293T cells were seeded in eight well ibidi dishes coated with poly-l lysine. Cells were transfected after 17 h using jetPrime (Polyplus). Therefore, 600 μg of total DNA was added to 25 μl JetPrime Buffer. DNA:jet Prime reagent ratio was 1:2. After 48 h cells were washed with

PBS and then fixed for 10 min at RT in 2% paraformaldehyde. Cells were once washed 2× with PBS. Cells were permeabilized with 0.5% Trition-X-100 in PBS for 5 min at RT and then washed 2× with PBS. Blocking solution (5% donkey serum in PBS/0.1% Tween-20) was added for 10 min at RT. First antibody (mouse anti-FLAG 1:500, 9A3, CellSignaling) in blocking solution was added overnight at 4 °C. Second antibody (donkey anti-mouse conjugated with Alexa Fluor™ 480, A-31570 Thermo Fisher Scientific) was added in blocking buffer for 1 h at RT. Cells were again washed 3× with PBS/0.1% Tween for 5–10 min and then stained with DAPI (0.5 µg/ml, 1:2000) in PBS for 5 min at RT. Afterwards, cells were washed 2× with PBS and kept at 4 °C until imaging. Cells were imaged using a LSM 900 microscope (Zeiss) using the Airyscan 2 setup. Images were processed using Fiji.

## RT-PCR
RNA from HEK293T cells transfected with gRNA/circ gRNA constructs was extracted using TRizol (15596026, Thermo Fisher Scientific) and treated with DNaseI before performing reverse transcription. RNA was reverse transcribed using gene-specific primers and Maxima -H reverse transcriptase (EP0751, Thermo Fisher Scientific). One microliter of cDNA was used for a second PCR (using inverse Primers) and products were run on a 1.8% agarose gel.

## Western blotting
$5.5 \times 10^5$ HEK293T cells were seeded in 6well plates and transfected roughly 16 h later with a total of 6ug DNA (1:1:1 ratio of plasmids encoding DKC1-constructs/reporters/gRNAs) using 12 µl jetPrime reagent (Polyplus) according to the manufacturer's instructions. Media was changed after 4 h and cells were grown for additional 48 h. Cells were harvested by adding 100 µl RIPA buffer (Chromotek) supplemented with cOmplete protease inhibitor (Roche), 1 mM PMSF, 1 mM MgCl$_2$, and SM nuclease directly onto the 6well dish. The dishes were incubated for 10 min on ice. Afterward, the lysate was collected, thoroughly resuspended, and centrifuged for 10 min at $20,000 \times g$ and 4 °C. Supernatant was collected and diluted with 150 µl dilution buffer supplemented with cOmplete and 1 mM PMSF. Next 12.5 µl equilibrated myc trap beads (Chromotek) were combined with 225 µl lysate (25 µl were aliquoted and used for loading control) and incubated for at least 1 h at 4 °C on a rolling shaker. Pulldown and wash steps were performed according to the manufacturer's instructions. Protein was eluted by adding 40 µl 2× leammli buffer (120 mM Tris/Cl pH 6.8, 20% glycerol, 4% SDS, 0.04% bromophenol blue, 10% β-mercaptoethanol) to beads and incubating for 10 min at 95 °C. Supernatant was then subjected to SDS-PAGE on a 4–12% Bis-Tris gel (NuPAGE, Thermo Fisher Scientific). Proteins were transferred to a nitrocellulose membrane (Trans-Blot Turbo Midi 0.2 µm Nitrocellulose Transfer Packs, Bio-Rad) using Trans-Blot Turbo Transfer System (Bio-Rad). Membranes were submerged in blocking buffer (5% milk powder in TBST) on a rolling shaker at RT for 1 h. The first antibody (anti-myc, 1:1000, Cell Signaling Technology, 9B11) was incubated at 4 °C overnight. Blots were washed three times 5–10 min with TBST on a rolling shaker. Next blots were incubated with second antibody, anti-mouse HRP (1:10,000, Jackson ImmunoResearch, 715-035-150) in blocking buffer for 1 h at RT. Before imaging blots were again washed four times 5–10 min with TBST and then incubated with 2 ml ECL Western Blotting Detection Reagent (Cytiva) for 5 min. Blots were imaged on a ChemiDoc imaging system (BioRad) imager and band intensities were analyzed using Fiji.

## RNA extraction
$1.5 \times 10^6$ HEK293T cells were seeded in P10 dishes and transfected 17–18 h later. For transfection, 48 µl of jetPrime reagent were mixed with 24 µg total DNA (plasmids were kept 1:1:1 ratio) in 1 ml jetPrime Buffer. Medium was exchanged after 4 h and cells were grown for 48 h. Cells were washed 1× with PBS and then trypsinized. After neutralization with DMEM, cells were collected in 15 ml centrifugation tubes. Cells were centrifuged for 5 min at $400 \times g$ at 4 °C. Supernatant was

poured off and cells were resuspended in 2 ml PBS. Cells were centrifuged again for 5 min at $400 \times g$ at 4 °C. Supernatant was poured off and the cell pellet was flash-frozen in liquid nitrogen and stored at −20 °C for further analysis. Poly-A RNA was isolated using mRNA Direct Kit (Invitrogen, 61012) according to the manufacturer's instructions. After two sequential rounds of oligo dT isolation poly-A RNA was eluted in 20 µl EB supplied with the kit, flash frozen in liquid nitrogen, and stored until later use. Eluted RNA was split and one half was used for DRS and the other for BID-seq.

## BID-seq
To assess the presence/absence of pseudouridine in oligo and polyA RNAs, a derived version of BID-Seq protocol[29] was performed. Briefly, 100 ng of RNA were subjected to RNA fragmentation followed by bisulfite treatment and desulphonation as described in ref. [30] Treated RNAs were end-repaired, purified, and subjected to NEBNext® small RNA library following manufacturer's protocol. The quality and quantity of each library were assessed using a high-sensitivity DNA Chip on a Bioanalyzer 2100 and a Qubit 2.0 fluorometer. High-throughput sequencing of the multiplexed libraries was performed on an Illumina NextSeq 2000 instrument in a 75 nt single-end read mode. Raw sequencing reads were inspected with FastQC and adapter sequence was removed by trimmomatic v0.39[31]. Alignment to the reference sequence was done by Bowtie2. v2.4.2[32] with "relaxed" alignment stringency, allowing to retain 1–3 nt gapped reads. Further analysis was done by samtools mpileup utility and counting the deletions at every position in the reference.

## Preparation and splinted ligation of RNA oligos
Custom 40 bp long RNA oligos (modified/unmodified at the targeted base) mimicking the target regions with our mCherry and eGFP reporters were ordered from Dharmacon. We ligated oligos in order to be long enough to be analyzed by DRS. RNA oligos were 5′-phosphorylated enabling them to be used in later ligations. Phosphorylation was performed using T4 Polynucleotide Kinase by NEB (M0201) following manufacturer instructions, incubating the reaction for 3 h @ 37 °C before terminating the reaction @ 65 °C for 20 min. Phosphorylated oligos were purified using Oligo Clean and Concentrator Kit by Zymo research (D4060) following manufacturer instructions. Purified oligos were ligated using T4 RNA Ligase 2 by NEB (M0239), combining equimolar amounts of oligos to be ligated, 1xT4 RNA Ligase Buffer, 10% PEG8000, 10 U T4 RNA Ligase 2 as well as a complementary cDNA splint ensuring correct order of ligation. Digestion of the cDNA splint was performed through DNase I by ThermoFisher Scientific (EN0525) following manufacturer's instructions and purified through Oligo Clean and Concentrator Kit. Poly(A) tailing of the purified ligation constructs was performed by Escherichia coli Poly(A) Polymerase by NEB (M0276) following manufacturer instructions and purified again using Oligo Clean and Concentrator Kit before proceeding to library preparation. Oligo1 consists of the modified target region of eGFP and the unmodified target region of mCherry. Oligo2 consists of the unmodified target region of eGFP and the modified target region of mCherry. Sequences are listed in Supplementary Table 5

## Direct RNA Nanopore-seq library preparation
Library preparation was performed using Direct RNA Sequencing Kit (ONT, SQK-RNA004) following the manufacture's protocol (DRS_9195_v4_revB_20Sep2023). Briefly, 100 ng of poly(A)-tailed RNA or poly(A)-tailed constructs were ligated to ONT RT Adaptor (RTA) using T4 DNA Ligase (NEB, M0202T) and was reverse transcribed using SuperScript III RT (Thermo Fisher Scientific, 18080044). The products were purified using 1.8X Agencourt RNAClean XP beads (Thermo Fisher Scientific, NC0068576) and RNA adapter (RLA) was ligated onto the RNA:DNA hybrid. For PolyA sample, the mix was purified using 0.6X Agencourt RNAClean XP beads whereas for the ligated construct

the mix was purified using 2× beads. Both were washed twice using wash buffer. The samples were then eluted in elution buffer (EB) and mixed with Sequencing Buffer and Library Solution before loading onto primed RNA chemistry PromethION flow cells.

### Preprocessing of direct RNA sequencing data

Raw direct RNA sequencing reads were base called using the Dorado base caller (v0.7.0) in super accuracy mode for direct RNA sequencing on RNA004 flowcells (Dorado: Oxford Nanopore Technologies, https://github.com/nanoporetech/dorado). Additionally, the recently integrated pseudouridine detection mode was executed. Subsequently, base-called reads were mapped onto the custom reference sequences from the two reporters (eGFP and mCherry) for the LCK-FUS[2–267]-MCP-DKC1, NES-DKC1, and Control samples and onto the custom oligo sequences for the synthetic oligo samples using minimap2 v2.28 with the suggested parameters for direct RNA reads: -ax splice -uf -k1424[33]. The resulting bam files were sorted, indexed, and filtered by samtools v1.16 using the -F 256 flag and specifying the regions of interest to be covered (Samtools: https://doi.org/10.1093/gigascience/giab008). Quality control of sequencing and mapping was done by NanoComp v1.23.125[34]. All sequencing metrics can be found in Supplementary Table 4.

### Pseudouridine detection using direct RNA sequencing reads

All subsequent analyses were executed in Python v3.10 using the filtered BAM files with the pysam package v0.22.1, a Python wrapper for HTSlib and Samtools (Pysam: https://github.com/pysam-developers/pysam)[35]. For accuracy metrics, the match and mismatch percentages for each reference base across all runs were extracted using Pysam's *pileup* function with parameters *max_depth = 10,000* and *min_base_quality = 13*[35] and plotted as heatmaps using the seaborn[35] v0.13.2 and matplotlib[36] v3.9.0 packages. All runs revealed a mean base accuracy >98%.

Previous studies on direct RNA sequencing have shown that pseudouridine modification sites can cause systematic basecalling errors, specifically uridine to cytosine mismatches using the older RNA002 flowcell. To verify whether this error persists with the newer RNA004 flowcell and chemistry, we sequenced two synthetic oligos containing identical 10-mer motifs to the target sites of the eGFP and mCherry reporters, with either 100% or 0% pseudouridine modification rates at one uridine position.

Using Pysam's pileup function with *max_depth = 100,000* to extract basecalling information, our positive control regions (100% modified) showed U > C mismatches in 88.3% of mapped bases for eGFP and 34.2% for mCherry specific motifs. The unmodified regions revealed U > C mismatches in only 3.9% and 1.3% of bases for eGFP and mCherry-specific motifs, respectively. These results indicate that despite advancements in direct RNA sequencing chemistry, flowcells, and basecalling models, the U > C mismatch remains indicative of pseudouridine modification, at least for the investigated motifs.

For reads that did not show a U > C mismatch, we used the newly developed Dorado pseU calling model to classify reads as modified or unmodified. Modification calling results from Dorado were extracted from the ML/MM tags in the BAM files, with probability values represented as $P = (p*256)\text{-}1$, where $p$ is the probability from 0.0 to 1.0. We used a probability threshold of $p > 0.95$ to extract only high-confidence sites. Combining reads with either a U > C mismatch or a Dorado classification, the synthetic oligo samples showed that the percentage of modified reads ranged between 96.6% and 91.8% for fully modified and 4.4% and 1.4% for completely unmodified sites of the eGFP and mCherry target motifs, respectively.

We then calculated the pseU percentage for both mCherry and eGFP motifs in LCK-FUS[2–267]-MCP-DKC1, NES-DKC1, and Control samples following the same approach. To remove standard error for specific sequence motifs in the pseudouridinylated samples, we subtracted the percentages of modified reads observed at the eGFP and mCherry target positions of the control sample from the pseudouridine percentages observed for LCK-FUS[2–267]-MCP-DKC1 and NES-DKC1 samples, respectively.

The U > C mismatch and Dorado-based modification percentages, both significant ($p < 0.95$) and insignificant ($p > 0.95$), were plotted for the sequence context −5 to +5 bases around the target positions of eGFP and mCherry in both the samples and synthetic oligos using ggplot2[37] v 3.4.4 in R v4.2.3 (Supplementary Figs. 7 and 8).

### Statistical analysis

In each flow cytometry measurement data from 50,000 live cells were collected. Median values were used for further analysis as they are more robust in transient transfections where population distributions are skewed. For each experiment, unless noted otherwise, three independent replicates were performed. All statistical analysis was done using Graph Pad Prism. We performed an unpaired two-sided *t*-test. We note that those tests are not defined for very low sample numbers and are thus used here instead as an indicator of substantial difference.

### Reporting summary

Further information on research design is available in the Nature Portfolio Reporting Summary linked to this article.

## Data availability

The sequencing data generated in this study have been deposited in the European Nucleotide Archive (ENA) at EMBL-EBI under accession code PRJEB76145. The raw data generated in this study are provided in the Source Data file. Source data are provided with this paper.

## Code availability

All scripts are available in the following GitHub repository: https://github.com/AnWiercze/Pseudouridine_detection_Schartel_et_al_2024.

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

## Acknowledgements

We thank the flow cytometry and microscopy core facilities of the IMB and JGU for their assistance. We thank all members of the Lemke lab for helpful discussions. We acknowledge funding of the ERC ADG Multi-OrganelleDesign and the Volkswagen Stiftung. We also acknowledge funding from the SFB1551 "Polymer concepts on Cellular function" of the Deutsche Forschungsgemeinschaft (DFG project number 464588647).

## Author contributions

L.S. and E.A.L. conceived the project. L.S. did flow cytometry, western blotting, and microscopy experiments, T.B. prepared the RNA library, S.M. prepared synthetic RNA oligos for sequencing, and A.W. performed bioinformatic analysis. C.J., M.H., and S.G. provided analytical tools and reagents. V.M. and Y.M. performed BID-seq. L.S. and E.A.L. co-wrote the manuscript.

## Funding

## Competing interests

L.S. and E.A.L. have filed a patent related to the manuscript. The remaining authors declare no competing interests.
