## [Transparent Peer Review file · Nature Communications]

Selective RNA pseudouridylation in situ by circular gRNAs in designer organelles

Corresponding Author: Professor Edward Lemke

Version 0:

Reviewer comments:

Reviewer #1

(Remarks to the Author)

Schartel et al. present an interesting study on the selective modification of RNAs by their colocalisation with pseudouridylation machinery. In this study, the authors use their previously designed recruitment fusions, that combine phase-separating domains and membrane-targeting motifs to create film-like organelles in HEK cells. Here, they use these organelles to selectively pseudouridylate mRNAs for targeted stop codon suppression.

Overall, the study is interesting, but needs clarification. The fusion proteins that the authors have constructed are complex, and I would urge the authors to be as clear as possible when describing them (and the components of their organelles). This study builds heavily on the authors' previous works. The distinguishing factor is the pseudouridylation. The authors show that their system can, through several iterations, improve the selectivity of RNA modifications, here shown by stop codon suppression. However, it's not clear what the eventual goal of this system is. It's hard to imagine this being used to treat genetic diseases caused by premature stop codons, given the complexity of the organelle system. It would be interesting to see this system developed further, for instance by evaluating splicing regulation, as suggested by the authors, or applying it to biologically relevant targets other than fluorescent reporters. In the past the authors have developed multiple orthogonal organelles, could a similar strategy be used to create multiple orthogonal RNA modifications? Overall, the authors have developed an interesting system, but one which could benefit from a little more development.

1. In Figure 1a the construct NES-DKC1 appears to have a punctate/non-diffuse localisation in the cytoplasm. Was this expected, and can the authors comment on how this might be affecting the efficiency of pseudouridylation? NES-DKC1 appears to be the best performing (though less specific) construct throughout. Is this accentuated by phase separation/localisation of cytoplasmic DKC1? Further, does this account for the relatively weak influence of phase-separating FUS on the organelle, in this arrangement?

2. Following, since there appears to be specific localisation of NES-DKC1, we would encourage the authors to image the other constructs tested in figure 1, rather than just the full-length and controls.

3. The authors investigation of circular gRNAs is interesting, though the overall improvement by circularisation is relatively meagre. The authors suggest that the improvement is driven by exonuclease resistance, yet they are lacking evidence for this hypothesis. Could this be tested to verify that the circularised RNAs are indeed persisting longer in the cytoplasm?

4. The authors mention in the discussion that only one of the four components of the DKC1 machinery were targeted to the organelle. This should be explicitly stated in the description of the design, where there is no mention of the other 3 components. Further, if this is a complex, it should be shown as such in the cartoon representations. This is useful to know when visualising what protein and nucleic acid components are present in the condensates. In general, it would be interesting to know how specific the organelles are in their components, i.e., what other proteins or complexes are non-specifically enriched in the organelles? This is something that has not been investigated by the authors, but is particularly important as the organelles are focused on providing specificity.

5. In the section on circular guide RNAs, the final paragraph (lines 129-139), the changes made to these constructs are again hard to visualise without diagrams or sequences, please provide some cartoons or schematics. Further, an additional FUS domain was added in these constructs, given that the improvement on circularisation is relatively meagre, could this be due to improved phase separation caused by duplication of the IDR?

6. Error bars are missing in several figures (1d, 3a, S1a, S2a). It looks like these have been omitted when values have been normalised to 1, but there will still be error on these normalised values, that should be represented. Further, there are no statistical analyses in the study throughout. These should be performed to demonstrate significance of the authors comparisons. In particular, the authors claim in lines 99-100 that some tested designs are different, that appear by eye to differ very marginally.

7. The fusion proteins and RNAs used in the study are clearly complex, and occasionally challenging to interpret. Providing a table of constructs and sequences for both in the SI would greatly help the clarity of the manuscript.

8. In particular, the different gRNAs used in supplementary figure 2 are challenging to visualise, as there are no sequences given or schematics of the different assemblies. Cartoons of the different gRNAs would be helpful here.

9. The authors mention the construct NES-DKC1, but there is no definition of what NES is (I presume this is nuclear export signal-DKC1, but this should be defined in the text). Further, is this NES present on all the constructs tested? If so, this should be represented.

Minor comments

• Different confocal images as well as the cartoons of the respective constructs are organized by rows, so in the caption to figure 1 they should be referred to as top and bottom rather than left and right.

It is our view that, in its current form, this study is interesting but lacks some novelty, and should be developed further before we can endorse publication in Nature Communications.

Reviewer #2

(Remarks to the Author)

Reviewer #3

(Remarks to the Author)

In this manuscript, the author has developed a film-like organelle architecture that facilitates selective pseudouridylation of target mRNA. This innovative approach combines specific proteins that undergo phase separation in cells with RNA editing enzymes, creating a unique phase separation environment. This methodology, previously validated by the authors for orthogonal protein translation, has now been extended to RNA editing. By utilizing the RNA-guided pseudouridine synthase dyskerin (DKC1), the system achieves RNA pseudouridylation, potentially offering new avenues for treating diseases through pseudouridine-mediated stop-codon suppression. Additionally, to overcome the instability of linear gRNA in RNA pseudouridylation, the efficiency has been enhanced through the introduction of circular gRNAs. While the manuscript's logic is sound, several concerns merit attention for the method to be deemed highly specific and reliable:

1. A major issue is quantifying mRNA pseudouridylation with this designer organelle system. Although Nanopore RNA-sequencing is employed for detection, its accuracy, particularly for RNA modification bases, is problematic. Alternative, locus-specific pseudouridine detection methods, such as qPCR-based assays (PMID: 28960747, PMID: 34020036), are recommended to validate RNA pseudouridylation success.

2. In Figure 1b, the fluorescence intensity of GFP in NES-DKC1 significantly surpasses that in LCK-FUS-MCP-DKC1. Assuming LCK-FUS-MCP-DKC1 more accurately reflects the film-like organelle architecture, this discrepancy raises questions. Did I make some mistakes about this result? In addition, the negative control without the MS2 aptamer should be added to prove the concept of specificity of this designer organelle.

3. For the fig 2d, the methodology for validation of gRNAs circularization is not clearly provided in supporting information or methods part. Using an exonuclease enzyme to digest linear RNA in purified PCR products could confirm circular RNA formation. Additionally, assessing whether the qPCR method is more apt for detecting circular RNA by targeting CircRNA's back-splicing sites would be beneficial.

Minor Concerns:

1. In the title, the RNA modification is too big. It should clarify the RNA pseudouridylation in the title.

2. A more detailed introduction of the design would allow readers to grasp the manuscript's core concepts without needing to consult the authors' previous publications. In addition, the scheme of the design is too simple to understand the main idea. Present figure is not good for the readers to understand this work quickly and easily.

Version 1:

Reviewer comments:

Reviewer #1

(Remarks to the Author)

We thank the authors for their response to our comments. However, some questions need to be addressed further before it can be endorsed for publication in Nature Communications. While the manuscript improved in clarity and the functional aspect, the mechanism in relation to the role of FUS and evidence of organelles needs to be demonstrated better.

There are several questions that arise with regards to the new images and the respective authors' responses.

1. Further clarifying on the subcellular localisation of fusion constructs, there is little difference in fluorescence distribution between any of the images provided in Fig. S1 and Fig. 2a. Though LCK seems to have improved the enrichment of the constructs at the plasma membrane (unclear from the images), there is still apparent localisation at the membrane even without the anchor. Is this a correct interpretation? If so, do the authors see it as a factor limiting the advantage of their membrane-anchored constructs in pseudouridylation efficiency?

2. Following, the aforementioned images have different contrast and brightness adjustments which make them harder to compare. They might be suggesting that FUS region allows for more favourable distribution of LCK-FUS-MCP-DKC1 and

LCK-FUS-DKC1 to the plasma membrane compared to LCK-MCP-DKC1, unless it's an artifact of image processing. Can the role of the IDR region be limited to a flexible linker enhancing accessibility of MCP and DKC1?

3. You state that LCK constructs missing either the IDR or MCP showed lower efficiency in comparison (Fig. 2b,c)

Following on the original comment 6 on the statistical analysis, with regards to the data presented in Fig. 2b and 2d, are there statistically significant differences demonstrating the role of FUS?

4. Taking into account the lack of phenotypical differences (comment 1) and little evidence of FUS region playing a role (comment 3), can the authors demonstrate the formation of film-like organelles, presumably driven by attractive interactions between the engineered protein constructs, as opposed to 'simple' immobilisation to the plasma membrane? The authors are using FUS region in the same manner as in previous studies, however, the new constructs are complex and drastically different and their assembly is determined by all components and might not be sufficiently driven by FUS to form a phase. Currently, LCK and also MCP, seem to play the major role in the observed efficiency/selectivity of pseudouridylation. At the same time, in the current form the role of FUS as well as phase separation and hence the existence of organelles remains elusive and speculative.

Reviewer #2

(Remarks to the Author)

The authors have been diligent in their revisions, and the revised manuscript is now much clearer and the constructs used are easier to follow. In particular the new Figure 1 is very helpful. There are still some interesting questions that stand out. In particular, the curious punctate phenotype of NES-DKC, and the potential recruitment of other components of the H/ACA complex. These are potentially relevant to understanding the phase behaviour and mechanisms of the H/ACA complex, and haven't been addressed here, but I accept that this is potentially out of scope of this study, which focuses on improving selectivity. Overall, the manuscript appears much improved.

Reviewer #3

(Remarks to the Author)

The authors have addressed all of my concerns, especially regarding the quantification of mRNA pseudouridylation by BID-seq. I have no further questions.

Responses in green

Reviewer #1 And Reviewer#2 (Remarks to the Author)

Schartel et al. present an interesting study on the selective modification of RNAs by their colocalisation with pseudouridylation machinery. In this study, the authors use their previously designed recruitment fusions, that combine phase-separating domains and membrane-targeting motifs to create film-like organelles in HEK cells. Here, they use these organelles to selectively pseudouridylate mRNAs for targeted stop codon suppression.

Overall, the study is interesting, but needs clarification. The fusion proteins that the authors have constructed are complex, and I would urge the authors to be as clear as possible when describing them (and the components of their organelles). This study builds heavily on the authors' previous works. The distinguishing factor is the pseudouridylation. The authors show that their system can, through several iterations, improve the selectivity of RNA modifications, here shown by stop codon suppression. However, it's not clear what the eventual goal of this system is. It's hard to imagine this being used to treat genetic diseases caused by premature stop codons, given the complexity of the organelle system. It would be interesting to see this system developed further, for instance by evaluating splicing regulation, as suggested by the authors, or applying it to biologically relevant targets other than fluorescent reporters. In the past the authors have developed multiple orthogonal organelles, could a similar strategy be used to create multiple orthogonal RNA modifications? Overall, the authors have developed an interesting system, but one which could benefit from a little more development.

We thank the reviewers for their comments. We have included an additional figure (now Figure1) showcasing the organelle concept for clarity. Our objective is to apply the organelle's selectivity principle for RNA modifications/editing, specifically using pseudouridylation as a demonstrative example. While acknowledging the complexity of the system, it represents a foundational step towards the development of novel RNA modification/editing tools with selectivity.

A major challenge in the RNA editing field are off targets. Our way of creating specialized compartments for sequestering reactions in "microreactors" point towards a totally new conceptual way of dealing with off targets and enhancing specificity. We show this now on a disease-relevant target gene by swapping mCherry to AldoB(148X), a gene mutated in hereditary fructose intolerance, where we can also show pronounced off target mitigation.

We would also like to stress that while we certainly benefitted from our experience on OTOs, the mechanisms of RNA editing and ribosomal translation are very different, as well as the expected challenges. Thus, showing that condensation concepts can help to lower off-target effects for RNA editing is a new approach that we feel is important and justified to be shared with the community.

1. In Figure 1a the construct NES-DKC1 appears to have a punctate/non-diffuse localisation in the cytoplasm. Was this expected, and can the authors comment on how this might be affecting the efficiency of pseudouridylation? NES-DKC1 appears to be the best performing (though less specific) construct throughout. Is this accentuated by phase separation/localisation of cytoplasmic DKC1? Further, does this account for the relatively weak influence of phase-separating FUS on the organelle, in this arrangement?

We have observed this punctate phenotype of NES-DKC1 as well but can only speculate on the influence of pseudouridylation efficiency. In general, one of the characteristics of the new film like organelle is a tradeoff in terms of efficiency towards specificity. Immobilization of DKC1 at our plasma membrane

organelle results therefore in an efficiency decrease contrary to "freely diffusing" NES-DKC1 in the cytosol, which was not surprising to us and something we have observed in other organelle arrangements for our orthogonally translating organelles^{1,2}. The influence of FUS-IDR is more critical in our organelle arrangement as it is in "non anchored" (constructs without LCK) fusion constructs. This might lead to an apparent puncta localization of NES-DKC1 on its own but is also purely speculative from our site at this point. For us, the images should more exemplify the differential localization of NES-DKC1 and the OREO, rather than make assumptions of the phase separating behavior, as this – in particular 2D phase separation is still understudied and also influenced by a variety of factors, such as even fixation of cells.

2. Following, since there appears to be specific localisation of NES-DKC1, we would encourage the authors to image the other constructs tested in figure 1, rather than just the full-length and controls.

We included confocal images of all measured constructs in Fig2b in FigS1.

3. The authors investigation of circular gRNAs is interesting, though the overall improvement by circularisation is relatively meagre. The authors suggest that the improvement is driven by exonuclease resistance, yet they are lacking evidence for this hypothesis. Could this be tested to verify that the circularised RNAs are indeed persisting longer in the cytoplasm?

We would respectfully disagree that the improvement by circularization is meagre with respect to reported state of the art in that field. The improvement is very relevant as pseudouridine mediated readthrough is not very efficient. Even small improvements in full length protein production can have a huge impact in diseases caused by premature stop codons. A recent paper has highlighted the significance of enhancing this process by employing near-cognate tRNAs to increase pseudouridine mediated readthrough³ by managing a similar, 1.5fold increase (for TAG stop codon) in eGFP+ cells on pre-gated transfected cells.

For the stabilizing effect of circularization of small RNAs we refer to previous works where circularization of gRNAs for A-I editing has been shown to stabilize and outlast linear gRNAs as well as to Litke et al who verified the enhanced stability on circular aptamers⁴⁻⁶.

4. The authors mention in the discussion that only one of the four components of the DKC1 machinery were targeted to the organelle. This should be explicitly stated in the description of the design, where there is no mention of the other 3 components. Further, if this is a complex, it should be shown as such in the cartoon representations. This is useful to know when visualising what protein and nucleic acid components are present in the condensates. In general, it would be interesting to know how specific the organelles are in their components, i.e., what other proteins or complexes are non-specifically enriched in the organelles? This is something that has not been investigated by the authors, but is particularly important as the organelles are focused on providing specificity.

Indeed, only the catalytic subunit of the H/ACA complex, DKC1 seems to be enough to modify our and also another system that was recently published⁷. The authors of this study also investigated if overexpression of any of the 3 other components could improve pseudouridylation but failed to do so. This is also in accordance with our results as we tried to place the other components of the complex in the organelle in an LCK-fusion construct but failed to observe an indirect increase in pseudouridylation in our selectivity assay (maintext Figure 2d). We explicitly state this now in the main text and also refer to Song et al. and our own observations. We also redraw Figure 1 with the tetrameric complex as suggested by the reviewer.

Regarding the investigation into the specificity of the organelle's components and the potential non-specific enrichment of other proteins or complexes within the organelles, we do agree that this is an interesting question, but beyond the scope of this paper which reports on a new concept of how to engineer specific function into mammalian cells. Studying the content of potential condensates is a major technical challenge, even for established membraneless organelles, as those cannot be easily purified or enriched, are in equilibrium with the rest of the cell, and transient interactions dominate the composition of the organelle.

5. In the section on circular guide RNAs, the final paragraph (lines 129-139), the changes made to these constructs are again hard to visualise without diagrams or sequences, please provide some cartoons or schematics. Further, an additional FUS domain was added in these constructs, given that the improvement on circularisation is relatively meagre, could this be due to improved phase separation caused by duplication of the IDR?

We apologize for potential confusion and have included additional cartoons of the ms2 tagged circular gRNAs in FigS3. The goal in this experiment was to show that additional tagging of gRNAs by means of incorporating ms2 stem loops can enrich them in the organelle in contrast to untagged gRNAs. In this arrangement we therefore used organelles that harbor 2 different RNA binding domains (MCP/4xλN22 peptide) enabling recruitment of boxb loop tagged reporter and ms2 loop tagged circ gRNAs. We separated the two RNA binding domains within the constructs with another FUS IDR as we speculated it would fulfill the purpose of a flexible linker and a potentiator for phase separation as pointed out by the reviewers.

6. Error bars are missing in several figures (1d, 3a, S1a, S2a). It looks like these have been omitted when values have been normalised to 1, but there will still be error on these normalised values, that should be represented. Further, there are no statistical analyses in the study throughout. These should be performed to demonstrate significance of the authors comparisons. In particular, the authors claim in lines 99-100 that some tested designs are different, that appear by eye to differ very marginally.

We apologize for the confusion and included error bars to normalized constructs in Fig1d, 3a, S1a, S2a as well as statistical tests, which confirm our previous conclusions.

We have removed the claims in line 99-100 since as the reviewers correctly point out the difference in this particular experiments is marginally/only a weak trend. This does not change anything about the other conclusions of the paper. We thank the reviewer for drawing our attention to this.

7. The fusion proteins and RNAs used in the study are clearly complex, and occasionally challenging to interpret. Providing a table of constructs and sequences for both in the SI would greatly help the clarity of the manuscript.

We now included tables with all fusion constructs, reporters and gRNAs used in the supplementary information

8. In particular, the different gRNAs used in supplementary figure 2 are challenging to visualise, as there are no sequences given or schematics of the different assemblies. Cartoons of the different gRNAs would be helpful here.

We included cartoons for gRNA in FigS2.

9. The authors mention the construct NES-DKC1, but there is no definition of what NES is (I presume this is nuclear export signal-DKC1, but this should be defined in the text). Further, is this NES present on all the constructs tested? If so, this should be represented.

We apologize for the confusion; NES is now clarified in the text as well as its presence in all tested constructs

Minor comments

- Different confocal images as well as the cartoons of the respective constructs are organized by rows, so in the caption to figure 1 they should be referred to as top and bottom rather than left and right.

We apologize for the confusion and refer to the confocal images to top and bottom as oppose to left and right.

Reviewer #3 (Remarks to the Author):

In this manuscript, the author has developed a film-like organelle architecture that facilitates selective pseudouridylation of target mRNA. This innovative approach combines specific proteins that undergo phase separation in cells with RNA editing enzymes, creating a unique phase separation environment. This methodology, previously validated by the authors for orthogonal protein translation, has now been extended to RNA editing. By utilizing the RNA-guided pseudouridine synthase dyskerin (DKC1), the system achieves RNA pseudouridylation, potentially offering new avenues for treating diseases through pseudouridine-mediated stop-codon suppression. Additionally, to overcome the instability of linear gRNA in RNA pseudouridylation, the efficiency has been enhanced through the introduction of circular gRNAs. While the manuscript's logic is sound, several concerns merit attention for the method to be deemed highly specific and reliable:

We thank the reviewer for the nice comments.

1. A major issue is quantifying mRNA pseudouridylation with this designer organelle system. Although Nanopore RNA-sequencing is employed for detection, its accuracy, particularly for RNA modification bases, is problematic. Alternative, locus-specific pseudouridine detection methods, such as qPCR-based assays (PMID: 28960747, PMID: 34020036), are recommended to validate RNA pseudouridylation success.

We now redid the whole nanopore analysis with the latest generation of low cells and additional controls. The manuscripts now presents data using RNA004 flow cells, which evidently surpass their processor (RNA002) previously employed for our RNA direct sequencing in terms of accuracy. We can show that this new flow cell technology allows for substantial improvements in accuracy over RNA002 as compared below. We also included synthetic modified RNA oligos mimicking our target regions within our mCherry and GFP reporters as analytical standards verifying the applicability of our direct RNA seq results.

In parallel we also now show new experiments using BID-seq^{8,9} (Bisulfite-induced deletion sequencing) as an orthogonal method to quantify pseudouridine levels at base resolution and can show that the overall trend for both methods is the same. We included two new experts on this methodology in the author list, who performed these experiments.

2. In Figure 1b, the fluorescence intensity of GFP in NES-DKC1 significantly surpasses that in LCK-FUS-MCP-DKC1. Assuming LCK-FUS-MCP-DKC1 more accurately reflects the film-like organelle architecture,

this discrepancy raises questions. Did I make some mistakes about this result? In addition, the negative control without the MS2 aptamer should be added to prove the concept of specificity of this designer organelles.

As mentioned above in response to reviewers 1&2, the spatial organelle concept comes with a tradeoff in efficiency towards specificity possibly attributed to the immobilization of DKC1 at the plasma membrane. This tradeoff in efficiency comes with a more than 3fold increase in selectivity which is strongly mitigated by employing circular gRNAs. Users can choose between both systems of course, but in our view this tradeoff is worthwhile as any type of RNA manipulation such as editing or modification, demands high specificity in order to achieve the goal of being a minimally invasive technology for future applications in basic science and potentially even medical applications.

3. For the fig 2d, the methodology for validation of gRNAs circularization is not clearly provided in supporting information or methods part. Using an exonuclease enzyme to digest linear RNA in purified PCR products could confirm circular RNA formation. Additionally, assessing whether the qPCR method is more apt for detecting circular RNA by targeting CircRNA's back-splicing sites would be beneficial.

We apologize for not being clear in explaining the assay employed to verify the circularity of our gRNAs. We chose a PCR verification method as employed by Yi et al. which only gives rise to a PCR product if the gRNA has been successfully circularized⁶. Here the inverse primer design (primers do not "face" each other) does only result in a PCR product if the gRNA is circular. We have expanded Fig2d with a cartoon explaining the assay and more clearly described it within the main text and methods.

Minor Concerns:

1. In the title, the RNA modification is too big. It should clarify the RNA pseudouridylation in the title.

We have changed the title to RNA pseudouridylation rather than RNA modification

2. A more detailed introduction of the design would allow readers to grasp the manuscript's core concepts without needing to consult the authors' previous publications. In addition, the scheme of the design is too simple to understand the main idea. Present figure is not good for the readers to understand this work quickly and easily.

As also mentioned by reviewer 1&2 we incorporated an additional representative Fig1 and additional cartoons in FigS4 to aid in understanding the manuscript and supplemented it with a detailed explanation in the main text.

Point by Point references

1. Reinkemeier, C. D., Girona, G. E. & Lemke, E. A. Designer membraneless organelles enable codon reassignment of selected mRNAs in eukaryotes. *Science* **363**, eaaw2644 (2019).
2. Reinkemeier, C. D. & Lemke, E. A. Dual film-like organelles enable spatial separation of orthogonal eukaryotic translation. *Cell* **184**, 4886-4903.e21 (2021).
3. Luo, N. *et al.* Near-cognate tRNAs increase the efficiency and precision of pseudouridine-mediated readthrough of premature termination codons. *Nat Biotechnol* (2024) doi:10.1038/s41587-024-02165-8.
4. Litke, J. L. & Jaffrey, S. R. Highly efficient expression of circular RNA aptamers in cells using autocatalytic transcripts. *Nat Biotechnol* **37**, 667–675 (2019).

5. Katrekar, D. *et al.* Efficient in vitro and in vivo RNA editing via recruitment of endogenous ADARs using circular guide RNAs. *Nat Biotechnol* **40**, 938–945 (2022).
6. Yi, Z. *et al.* Engineered circular ADAR-recruiting RNAs increase the efficiency and fidelity of RNA editing in vitro and in vivo. *Nat Biotechnol* **40**, 946–955 (2022).
7. Song, J. *et al.* CRISPR-free, programmable RNA pseudouridylation to suppress premature termination codons. *Molecular Cell* **83**, 139-155.e9 (2023).
8. Dai, Q. *et al.* Quantitative sequencing using BID-seq uncovers abundant pseudouridines in mammalian mRNA at base resolution. *Nat Biotechnol* 1–11 (2022) doi:10.1038/s41587-022-01505-w.
9. Zhang, L.-S. *et al.* BID-seq for transcriptome-wide quantitative sequencing of mRNA pseudouridine at base resolution. *Nat Protoc* **19**, 517–538 (2024).

Point by point to revision 1

Answers to reviewers comments in green

Reviewer #1 (Remarks to the Author):

We thank the authors for their response to our comments. However, some questions need to be addressed further before it can be endorsed for publication in Nature Communications. While the manuscript improved in clarity and the functional aspect, the mechanism in relation to the role of FUS and evidence of organelles needs to be demonstrated better.

There are several questions that arise with regards to the new images and the respective authors' responses.

1. Further clarifying on the subcellular localisation of fusion constructs, there is little difference in fluorescence distribution between any of the images provided in Fig. S1 and Fig. 2a. Though LCK seems to have improved the enrichment of the constructs at the plasma membrane (unclear from the images), there is still apparent localisation at the membrane even without the anchor. Is this a correct interpretation? If so, do the authors see it as a factor limiting the advantage of their membrane-anchored constructs in pseudouridylation efficiency?

The reviewer has a good point, and we thank him for drawing our attention to this. Indeed, the relatively small volume of HEK cell cytoplasm made it hard to distinguish the differences with the chosen image settings. We now reimaged with higher resolution (airyscan) and also show zoom images in Figure 1. Here, one can clearly see that constructs harboring LCK have a distinct localization at the plasma membrane, while without LCK, the fluorescence appears dotted but equally distributed in the cytosol (updated Fig. 1a, new FigS1 in manuscript).

2. Following, the aforementioned images have different contrast and brightness adjustments which make them harder to compare. They might be suggesting that FUS region allows for more favourable distribution of LCK-FUS-MCP-DKC1 and LCK-FUS-DKC1 to the plasma membrane compared to LCK-MCP-DKC1, unless it's an artifact of image processing. Can the role of the IDR region be limited to a flexible linker enhancing accessibility of MCP and DKC1?

We apologize for any confusion and now provide new images with brightness and contrast adjusted equally between each other (Fig2a, Supplementary Fig.S1). We respond to the comment of FUS acting as a flexible linker under the next point.

3. You state that LCK constructs missing either the IDR or MCP showed lower efficiency in comparison (Fig. 2b,c)

Following on the original comment 6 on the statistical analysis, with regards to the data presented in Fig. 2b and 2d, are there statistically significant differences demonstrating the role of FUS?

4. Taking into account the lack of phenotypical differences (comment 1) and little evidence of FUS region playing a role (comment 3), can the authors demonstrate the formation of film-like organelles, presumably driven by attractive interactions between the engineered protein constructs, as opposed to 'simple' immobilization to the plasma membrane? The authors are using FUS region in the same manner as in previous studies, however, the new constructs are complex and drastically different and their assembly is determined by all components and might not be sufficiently driven by FUS to form a phase.

Currently, LCK and also MCP, seem to play the major role in the observed efficiency/selectivity of pseudouridylation. At the same time, in the current form the role of FUS as well as phase separation and hence the existence of organelles remains elusive and speculative.

We reply to point 3 and 4 together. In our introduction we state:

“Here, we demonstrate that a film-like organelle architecture enables mRNA-selective pseudouridylation as a proof-of-principle for a particular mRNA-modifying organelle. The concept behind film-like organelles is to create a unique biochemical environment within a living mammalian cell by borrowing principles from 2D phase separation^{11,12}. Primarily, mRNAs targeted at the thin film should be selectively modified, whereas mRNA located elsewhere in the cytoplasm should get less frequently modified. As the environment of the thin-film is in equilibrium with the cytoplasm, there are no physical boundaries to be passed, which abrogates the need for complex transport machinery. Furthermore, components can be shared between the host cell and the designer organelle so that film-like organelles can be simple in design and built from a few critical components while borrowing others from the cytoplasm.”

“

What we show is that spatial targeting to the plasma membrane gives a statistically significant specificity gain. We show functional evidence that our system works, and we refer to this as an OREO, as such the concept of an organelle is defined by us in terms of specific function. We acknowledge, that other definitions of organelle might be possible in biology, such as morphological appearance (e.g. well applicable for membrane encapsulated organelles), but we chose the functional definition. Additionally, as we clearly show, the best selectivity is obtained by membrane association, and such a structure can arguably be considered an organelle from a morphological perspective. However, we do acknowledge the semantic complexities and thus adhere to the functional definition.

Additionally, we now screened different organelles, based on different membrane anchors (new FigsS4 and FigS5 in manuscript). We utilized the same membrane anchors from our 2021 study in *Cell*, where we found that FUS (full length) improved selectivity in 3 out of 4 organelle constructs. While not essential for achieving the desired selectivity enhancement, FUS LCD now consistently proved beneficial enhancement in all our RNA modification organelles.

The concept of 2D phase separation is indeed very complex, and extensively under scrutiny and debate in the current literature, and as such we also refer in our paper that our methodology as inspired by the concept of phase separation, which enables easy exchange of biomolecules with cytoplasm and vs. In 2D phase separation, the effective concentration is exceedingly high. The design of an inert linker is not trivial, as at very high concentration, even PEG can stick to each other and phase separate. Thus, there is no easy way to distinguish such effects, and if one takes dynamics and potential heterogeneity in the membrane layer into account, things get even more complex, and thus this topic is currently the focus of many theoretical studies^{1,2}. Even DKC1 might become associative under such high concentrations. Our study is not aimed to decipher the biophysical mechanism, but to show a new bioengineering tool to enhance the precision of RNA editing. Specificity enhancement is obtained by spatial targeting of the enzyme and the target RNA together, and this we define as a new functional organelle as very common in synthetic biology.

Reviewer #2 (Remarks to the Author):

The authors have been diligent in their revisions, and the revised manuscript is now much clearer and the constructs used are easier to follow. In particular the new Figure 1 is very helpful. There are still some interesting questions that stand out. In particular, the curious punctate phenotype of NES-DKC, and the potential recruitment of other components of the H/ACA complex. These are potentially relevant to understanding the phase behavior and mechanisms of the H/ACA complex, and haven't been addressed here, but I accept that this is potentially out of scope of this study, which focuses on improving selectivity. Overall, the manuscript appears much improved.

We are thankful of the acknowledged improvements. We also appreciate that the reviewer acknowledges that our study focuses on improving selectivity, and that an in-depth biophysical study is out of scope, in particular for something as complex as 2D spatial targeting.

Reviewer #3 (Remarks to the Author):

The authors have addressed all of my concerns, especially regarding the quantification of mRNA pseudouridylation by BID-seq. I have no further questions.

We are thankful of the acknowledged improvements.

Point by Point References

1. Zhao, X., Bartolucci, G., Honigmann, A., Jülicher, F. & Weber, C. A. Thermodynamics of wetting, prewetting and surface phase transitions with surface binding. *New J. Phys.* **23**, 123003 (2021).
2. Jülicher, F. & Weber, C. A. Droplet Physics and Intracellular Phase Separation. *Annu. Rev. Condens.*

Matter Phys. **15**, 237–261 (2024).